ecology/environmental science/ecosystems

Brazilian Atlantic forest, conservation on private lands, ecotourism, functional connectivity, ecological corridors, ecological restoration

**Author for correspondence:**
Carlos E. V. Grelle
e-mail: cevgrelle@gmail.com

# Sustainability issues in a tropical mega trail

Carlos E. V. Grelle[1], Julia Niemeyer[2], Ernesto B. Viveiros de Castro[3,4], André M. Lanna[2], Mariella Uzeda[5] and Marcus Vinicius Vieira[1]

[1]Department of Ecology, and [2]Graduate Program in Ecology, Federal University of Rio de Janeiro, Rio de Janeiro, RJ, Brazil
[3]Serra dos Orgaos National Park, Chico Mendes Institute for Biodiversity Conservation, Teresópolis, RJ, Brazil
[4]School of Natural Resources and Environments, University of Florida, Gainesville, FL, USA
[5]Center of Agrobiology, Brazilian Agricultural Research Corporation, Brasilia, Brazil

CEVG, 0000-0002-8586-8655; JN, 0000-0003-2581-7255;
EBVdC, 0000-0002-1564-3071; AML, 0000-0003-4996-4399;
MU, 0000-0002-8675-3104; MVV, 0000-0002-4472-5447

Sustainability is a target that involves many socio-ecological questions, depends on opportunities and combines different initiatives. This can be especially difficult in regions with high biodiversity scores, mega cities, high level of human populations and an intense and long-standing land use. Here, we show how a mega trail, named Atlantic Forest Trail, within the Brazilian Atlantic Forest can join the protection of biodiversity and sustainable tourism through a 4270 km corridor connecting protected areas and crossing a variety of landscapes. Further, we show some initiatives of ongoing biodiversity monitoring, and an analysis of ecological restoration in private lands that can be applied in many regions to improve habitat connectivity for both biodiversity and human use.

## 1. Introduction

The diversification of tropical biodiversity, and its variety of landscapes has caused enchantment in European naturalists since the eighteenth century. With around $1\,200\,000\,km^2$ of extension, the Atlantic Forest harbours an impressive number of species, and is certainly one of the most diverse biomes in the world. The high species diversity is determined by the great variation of habitats in all its geographic extension [1], which in the Brazilian Atlantic Forest includes woody and herbaceous vegetation formations, distributed among a variety of reliefs and landscapes [2]. Since in the Brazilian Atlantic Forest there are landscapes from beaches to mountains [3], it is also possible to find a climatic gradient of 10°C within 100 km only, as in the Rio Janeiro State for example.

However, the Brazilian Atlantic Forest has been explored since the sixteenth century. Land use and land cover change increased over the last 70 years [4,5] bringing consequences to plants [6] and vertebrates species [7–9]. Consequently, many conservation initiatives were developed, including the creation of national and state protected areas (categories I–VI of IUCN), private protected areas (RPPNs in Portuguese acronym), corridors and action plans for protected species [10–14]. Besides being the first place to be occupied and explored by Europeans, the Atlantic Forest is home of at least 150 million people and is the main industrial centre, responsible for 80% of Brazil's gross domestic product (GDP) [15]. Because of that, the biome is one of the most deforested in Brazil, with the remaining vegetation cover comprising 13% of the original extent [16], and it is recognized as a world biodiversity hotspot [17]. Therefore, the Brazilian Atlantic Forest has complex socio-ecological questions, and conservation initiatives should include biodiversity and socioeconomic targets [18–21].

About 80% of the remaining Atlantic Forest vegetation has less than 50 ha and is inside private lands [22], thus forest restoration within private lands is essential to reconnect protected areas and improve conservation outcomes [23]. With that in mind, the Brazilian Native Vegetation Protection Law (Law No. 12.651/2012) determines how much native vegetation must be conserved and/or restored in specific areas called legal reserves (LR) and permanent protection areas (PPAs) within private lands in Brazil. The amount of forest depends on the size of a property (measured in fiscal modules) and the biome in which it is located. The fiscal module is an agrarian measure used in Brazil, fixed by county and expressed in hectares. It is calculated based on a county's main exploration type and income, corresponding to the minimum area necessary for a rural property to have an economic viable exploration [24]. In the case of the Atlantic Forest, properties larger than four fiscal modules must conserve or restore LR in 20% of the property's size, from which forested areas in PPAs can be subtracted. All properties must conserve and/or restore vegetation in PPAs, such as river margins, areas around lakes and water bodies, areas with slope higher than 45° and mountain tops [24].

While restoration in LR maintains a minimum habitat amount for biodiversity, vegetation in PPAs may serve as stepping stones and corridors to allow species to move between forest fragments, increasing gene flux and reducing local extinctions [25]. Therefore, land-use change within private lands has a key role for the conservation of ecosystem services and biodiversity. Niemeyer *et al.* [23] showed that forest restoration inside private lands in the Atlantic Forest has the potential to increase landscape permeability which, complemented by the conservation of remaining forested areas, contributes to biodiversity conservation in the long term.

Due to the already cited variety of landscapes, the Atlantic Forest is a biome with great touristic potential. Inspired in the North-American Appalachian Trail [26], in 2012 was launched the idea of implementing a mega trail crossing the Serra do Mar and Serra Geral mountain ranges. Mega trails are routes that cover many thousands of kilometres, crossing different states, regions or countries [27]. The Atlantic Forest Trail (AFT) is 4270 km long on its main axis, crossing 130 protected areas (figure 1 and electronic supplemental material), and the north to south orientation makes this a potential corridor for biodiversity in response to climate change [3,28]. The AFT was defined in many workshops with protected area managers, climbing and hiking organizations, tour operators and NGOs [3]. The project has four main goals: trail implementation and consolidation of protected areas; civil society engagement in conservation; promotion of green business, strengthening the ecotourism value chain; and the increase of the connectivity among the remnants of this biome through forest restoration and/or friendly land use (e.g. agroforest). However, this last goal depends on an area larger than a hiking trail, and a buffer is needed to act as a corridor of biodiversity.

To advance in this point above, here we plot a buffer of 4 km in the AFT (2 km in each side of the trail). Recently, Beier [29] suggested a minimum width of 2 km, but his own figure 1 shows an asymptote of complete stability, between the percentage of terrestrial mammal home-range that fit in a corridor, with 4 km of width. We estimate the amount of forest cover inside this buffer and discuss it in the light of habitat amount hypothesis. In addition, we compile and map the ongoing initiatives of tourism and biodiversity monitoring that can serve as a reference for the entire trail corridor and as a pilot for a monitoring network. Further, we selected the most degraded region to a more detailed analysis, estimating the environmental debts and restoration opportunities within private lands around the trail, following the legal requirements of the Brazilian Native Vegetation Protection Law, as already described above.

## 2. Methods

The AFT is 4270 km long, if we consider only the central line, discarding secondary branches to access tourism attractions and scenic views. Excluding sections over the sea, the trail extension is 3906 km.

R. Soc. Open Sci. 8: 201840

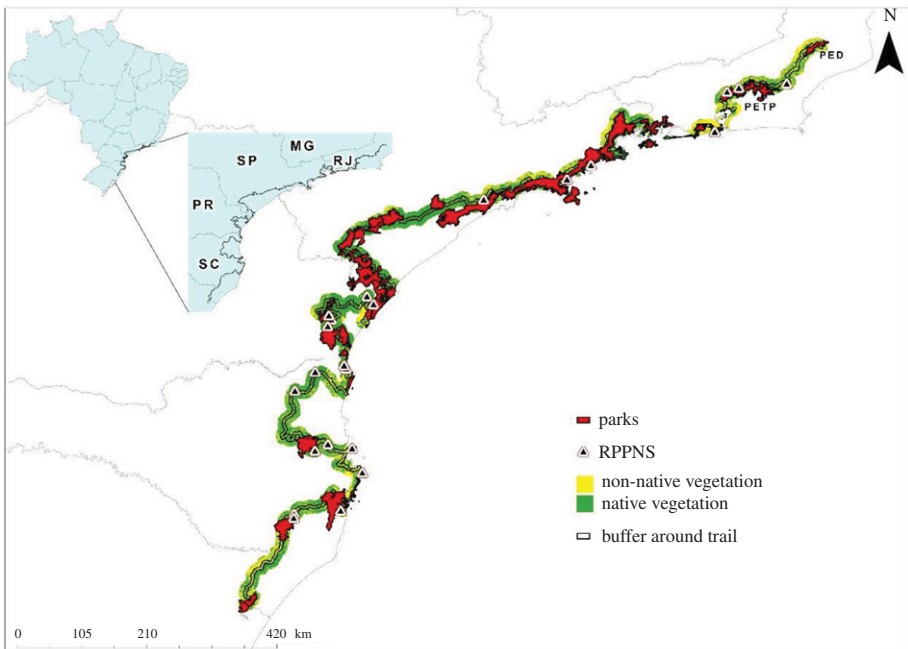

**Figure 1.** Atlantic Forest Trail (AFT) in Brazil, with 4 km buffer (2 km of each side) and private (RPPNs) and public parks along the trail, with spotlight of two state parks (Três Picos and Desengano) in the end of the trail.

During touristic activities, boats will be used to travel sea when it necessary. The trail crosses 130 protected areas, including 10 National Parks and other federal, state, local and private lands. The route passes through the mosaic of evergreen forest (mostly along the coast), semideciduous forest, deciduous forest, mixed forest (*Araucaria* forest), mangroves and 'restingas' that forms the Atlantic Forest. Oliveira-Filho & Fontes [30] described a gradient of species composition, but definitively stated that all these phytophysonomies formed a unique biome known as Atlantic Forest.

We plotted a buffer with 2 km (each side) in AFT shapefile using ArcGIS 10.5 [31], and crossed it with a land use and land cover map of the Atlantic Forest produced by the MapBiomas project (v. 3.0, 2018) to estimate forest cover (%) and altered areas. The resolution is 30 m, and the MapBiomas project give legends to native forest, pasturelands, mosaic of agropastoral land uses, monoculture tree plantation and croplands. To carry out the analyses, we clipped the land use land cover (LULC) map inside this buffer and selected only the classes related to native vegetation (corresponding to Forested Areas and Mangroves classes in MapBiomas map). Before the analyses, we excluded trail sections over the sea. The extension inside and around protected areas was also calculated. In order to analyse restoration opportunities in more detail, we selected the section between the Rio de Janeiro State Parks of Desengano and Três Picos, since this is the most deforested area along the AFT. The area surrounding these State Parks has a history of degradation and fragmentation due to agriculture expansion and urbanization [32]. Forest restoration within private lands is mentioned in the management plans of both state parks as a highly important action for conservation [33,34]. The counties that correspond to the area between the two state parks are Bom Jardim, Nova Friburgo, Macaé, Trajano de Moraes e Santa Maria Madalena. While Nova Friburgo and Macaé have industrial activity and important tourist attractions, the other municipalities are among the poorest in the State of Rio de Janeiro.

In the surrounding areas, most agriculture is dedicated to self-consumption or small local markets and is not profitable. In some regions such as the Santa Maria Madalena county, the lack of job opportunities and poverty has led to a rural exodus by the young generation [35]. Local private landowners may benefit from forest restoration, conservation and the ecotourism related to the AFT by conserving and managing the area, while complying with the law.

We collected the shapefiles of Desengano and Três Picos State Parks from the Instituto Chico Mendes (ICMBio/MMA), and the shapefile of the AFT drawn by coauthor E.B.V. Castro to this article. We obtained a LULC map of the Atlantic Forest from the MapBiomas project (v. 3.0, 2018). We obtained a map of the counties geopolitical divisions from the Instituto Brasileiro de Geografia e Estatística (IBGE). And finally, we obtained a map of the boundaries of the counties' private rural properties and their PPAs from the Sistema Nacional de Cadastro Ambiental Rural (SiCAR; http://www.car.gov.br/).

We selected the private properties of these counties that touched the boundaries of the 4 km buffer (2 km each side). From the LULC map, we selected only the classes related to native vegetation (corresponding to Forested Areas and Mangroves classes in the MapBiomas map).

We calculated the percentage of native vegetation inside the 2 km buffer and the environmental debt of each property following the Brazilian Native Vegetation Protection Law (Law No. 12.651/2012). This law determines the amount of native vegetation that landowners must restore and protect inside their land, in areas called LR and PPAs. Properties are classified based on their size in fiscal modules. All properties must restore and/or conserve native vegetation inside specific PPA areas, but only properties larger than or equal to four fiscal modules must restore LR. In the case of the Atlantic Forest biome, LR must cover 20% of the property's size, from which vegetated PPA could be subtracted [24].

Following that, we first calculated the total amount and percentage of forest inside all properties and their PPAs (if they exist). For properties larger than four fiscal modules, the percentage LR area that must be restored was calculated as

$$\%\mathrm{LR} = 20\% - \%\mathrm{FOR},$$

where %LR is the percentage legal reserve area that must be restored and %FOR is the percentage of forest inside the property.

To calculate a property's total debt, we subtracted the amount of forest inside PPAs from the LR area, and added the debt inside PPAs, which must also be restored. The property's total environmental debt was calculated as

$$\mathrm{Debt}\,(\mathrm{m}^2) = \%\mathrm{LR} * \mathrm{AREA}_{\mathrm{prop}} - \mathrm{FOR}_{\mathrm{PPA}} + \mathrm{Debt}_{\mathrm{PPA}},$$

where %LR is the percentage legal reserve area that must be restored, $\mathrm{AREA}_{\mathrm{prop}}$ is the property area (in $\mathrm{m}^2$), $\mathrm{FOR}_{\mathrm{APP}}$ is the forested area inside PPA (in $\mathrm{km}^2$) and $\mathrm{Debt}_{\mathrm{PPA}}$ is the area inside PPA that must be restored (in $\mathrm{m}^2$). For all the properties smaller than four fiscal modules, the environmental debt inside LR is 0, and the total environmental debt is equal to the area inside PPA that must be restored. All analyses were made using ArcGIS 10.5 [31].

Projects related to monitoring and reintroduction of species in the AFT region that is the AFT plus a buffer with 4 km of width were mapped. In our search, we included monitoring initiatives that are being developed over time, and thus inventories initiatives were not included. This was not meant to be an exhaustive research, but a selection of initiatives that can be replicated and implemented in other regions of the biome.

The survey of best practices in local and sustainable tourism was carried out through queries with the network of partners and volunteers engaged in the trail implementation. The cited examples were selected by its respect to sustainability principles and their potential for replication in other sections of the trail, also seeking to represent the diversity of initiatives, including inter-municipal tourist circuits, rural tourism and community-based tourism. Since parks have as basic objective precisely to make biodiversity conservation compatible with tourism and recreation [36], they were also identified on the map to highlight the potential of ecotourism throughout AFT.

## 3. Results

We found a forest cover of 75% in the 2 km buffer around the AFT (figure 1). About 55% (2160 km) of the trail are located inside protected areas and other 21% are less than 2 km from the limit of a protected area, therefore, being part of the corridor considered in this study and totalizing 76% of the trail extension inside or in the buffer zone of protected areas.

In the region between the Desengano and Três Picos State Parks there is a total of 478 rural properties that touch the buffer in the area between the two State Parks (figure 2), from which 299 are smaller than four fiscal modules and 381 (approx. 80%) have environmental debt, hence must perform forest restoration to comply with the law. Of those, only six properties must restore LR. In total, the region has the potential of restoring 17 317.50 $\mathrm{km}^2$ of Atlantic Forest.

Initiatives to monitor and manage biodiversity are distributed along the whole trail corridor (table 1 and figure 3). We mapped research projects working with large and medium mammals. Among the initiatives that have been developed throughout these large forest continuums of the AFT, two main types are going to be contemplated in this study: biodiversity monitoring and reintroduction of medium and large mammals. Monitoring includes research that seeks to understand how species are distributed and what are the processes related to their distribution retraction and expansion along the

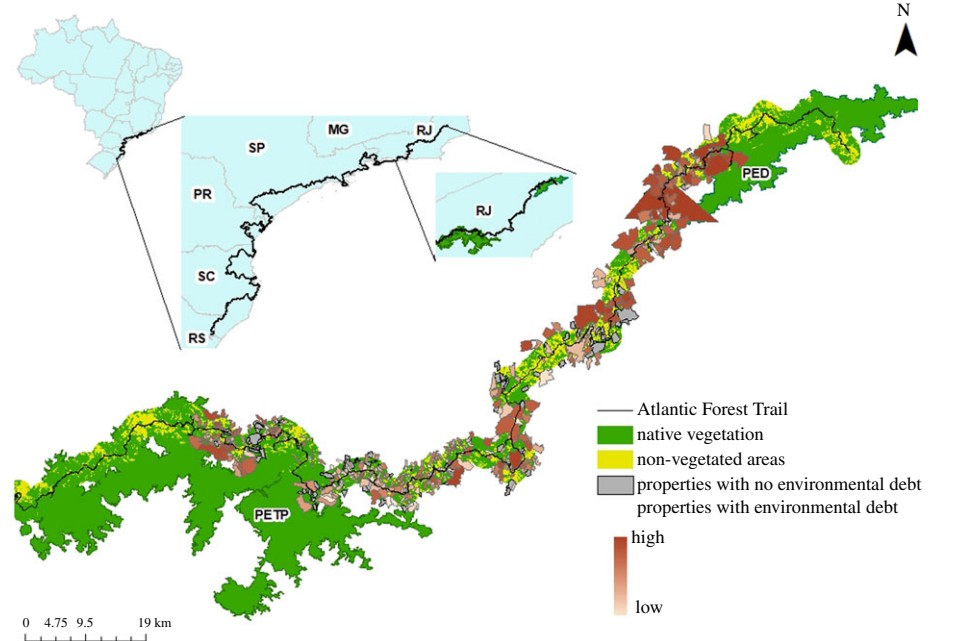

**Figure 2.** Land cover between Desengano and Três Picos State Parks in the Atlantic Forest Trail in Brazil, rural properties, and its environment debt due the lack of forest and the necessity of restoration. See text to read about details on environment debt.

**Table 1.** Current biodiversity monitoring projects in the Atlantic Forest Trail in Brazil. RS, Rio Grande do Sul State; SC, Santa Catarina State; PR, Paraná State; SP, São Paulo State; RJ, Rio de Janeiro State.

| project | state | reference |
|---|---|---|
| Serra dos Órgãos National Park | RJ | Aximoff *et al.* [37] |
| Federal University of Rio de Janeiro | RJ | Lanna [38] |
| Atlantic Forest Trail | RJ, SP, PR, SC, RS | http://caminhodamataatlantica.org.br/ |
| Refauna | RJ | Cid *et al.* [39] |
| REGUA | RJ | https://refauna.wixsite.com/site/projeto-anta |
| Muriquis da Bocaina | RJ, SP | Eurico [40] |
| Grandes mamíferos da Serra do Mar | PR | http://www.institutomanaca.org.br/ |
| Instituto Pró-muriqui | SP | Coles *et al.* [41] |

biodiversity corridor [42,43]. Given the great extent of the AFT and the biodiversity corridor, the restoration of connectivity would bring medium- and long-term responses. Meanwhile, the reintroduction of species brings quicker responses, with the re-establishment of populations and ecosystem services [44,45]. A prominent initiative is the Refauna Project, which has been working to reintroduce species such as the tapir, the largest terrestrial mammal in Brazil, in two stretches of the AFT [39]. These actions make it possible to anticipate the return of species that have been locally extinct and already indicate other species that could also be reintroduced, such as the peccaries [46].

Looking at tourism, the survey of best practices in sustainable tourism identified 13 initiatives that represent the modalities with the greatest potential for dissemination along the trail, with three circuits promoted by municipalities consortia and tour operators; seven community-based tourism projects; two rural tourism initiatives; and one conservation project that promotes multiple tourist activities (table 2). In figure 3, we point out some examples of different tourism initiatives along the AFT that show the potential to associate ecotourism with efforts to promote functional connectivity among the remnants of the Atlantic Forest in Serra do Mar. In the survey of the parks along the AFT, 10 National, 29 State and 17 Municipal parks were mapped, covering 1315 km or 33% of the total length of the mega trail (figure 3).

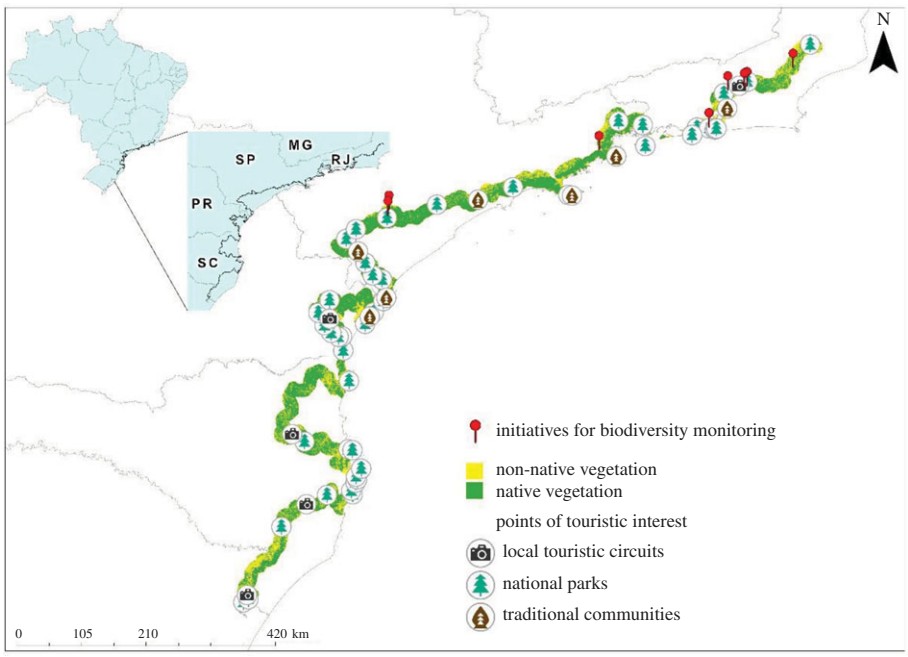

**Figure 3.** Initiatives of biodiversity monitoring projects and points of ecotourism along the Atlantic Forest Trail in Brazil.

**Table 2.** Best practices initiatives in local tourism along the Atlantic Forest Trail in Brazil. RS, Rio Grande do Sul State; SC, Santa Catarina State; PR, Paraná State; SP, São Paulo State; RJ, Rio de Janeiro State.

| name | type | state | source | reference |
|---|---|---|---|---|
| Pathways of the Southern Canyons Geopark | regional route | RS-SC | https://canionsdosul.org/ | |
| Acolhida na Colonia | rural tourism | SC | https://acolhida.com.br/ | Guzzatti et al. [47] |
| European Valley | regional route | SC | https://circuitovaleeuropeu.com.br/ | Pedrini et al. [48] |
| Gates of Atlantic Forest Great Reserve | conservation project | SC-PR-SP | http://grandereservamataatlantica.com.br/ | |
| Quilombos of Ribeira River Valley | CBT | SP | https://www.quilombosdoribeira.org.br/ | Rabinovici [49] |
| Tere-Fri Circuit | rural tourism | RJ | https://turismo.teresopolis.rj.gov.br/onde-ir/circuito-tere-fri/ | D'Onofre [50] |
| Caiçara Community of Barra do Superagui | CBT | PR | | Camargo [51] |
| Caiçara Community of Marujá | CBT | SP | http://garupa.org.br/guia-garupa/maruja-ilha-do-cardoso/ | Klimke et al. [52] |
| Tenondé-Porã Indigenous Land | CBT | SP | https://tenondepora.org.br/ | |
| Caiçara Community of Castelhanos | CBT | SP | https://www.castelhanos.org/ | |
| Nhandereko Network | CBT | SP-RJ | https://www.otss.org.br/turismo-de-base-comunitaria | Buck [53] |
| Nós da Guanabara Network | CBT | RJ | https://redenosdaguanabara.com.br | |

# 4. Discussion

Although the Atlantic Forest has only 13% of its original area covered by native vegetation, the corridor along the AFT is relatively well maintained. Habitat amount in landscapes has emerged as the most important determinant of species richness of all groups of organisms worldwide [54], particularly in the Atlantic Forest of South America [55,56]. Forest cover may be a proxy for habitat amount at large spatial scales, such as the whole Atlantic Forest biome, even though at local spatial scales this assumption may not hold [57]. Initiatives such as the AFT will favour the preservation and restoration of continuous forest remnants throughout the main axis of this coastal biome, increasing or at least maintaining forest cover. Higher species richness are also frequently associated with centres of endemism and evolutionary cradles, whose connection may also be secured by the AFT.

The central region of AFT, in the middle of Serra do Mar Corridor, is where the largest fragments and almost all species with historical occurrence are located [58]. Habitat loss and fragmentation were more intense in the Serra do Mar extremities. Local extinctions follow the same pattern, especially of larger species such as the jaguar, the tapir and the peccaries [58]. Seeking yet to conserve biodiversity, and considering the proximity of the trail to large urban and research centres, several initiatives for biodiversity monitoring are being developed. These aim at finding ways for a more sustainable relationship between fauna and human, such as the identification of priority altitudes for the restoration of connectivity. The various research and monitoring projects are gradually integrating and the engagement of trail volunteers in operational support can facilitate the implementation of a monitoring network along the entire trail, as has already happened on the Appalachian Trail [26].

Connectivity conservation is already recognized as an important, central issue in Conservation Science and Landscape Ecology [59], and for sustainable landscapes through the maintenance of ecosystem services [60]. Connectivity among forest remnants is equally important to maintain biodiversity, at large geographical scales [61], and within the Atlantic Forest biome [62,63]. Connectivity initiatives in landscapes are traditionally viewed as establishing corridors that connect habitat remnants, a structural connectivity in the sense that a landscape structure (corridors) is physically connecting forest remnants [64]. Corridors of native vegetation may be important to connect trails along the AFT, using the 2 km buffer as a minimum width of corridors (figure 1). Organisms are not able to cross the distance between forest remnants at large spatial scales, once these distances are larger than the movement abilities of individual organisms [19,65].

Other alternative strategies to promote functional connectivity and sustainable land use involve the establishment or simply the preservation of stepping stones, small patches of forest, or even scattered trees in pastures, that many forest organisms use as temporary shelters, or as beacons to orient themselves in a sea of grass. The importance and influence of these matrix elements on biodiversity are being described as countryside biogeography [66]. Scattered trees in pastures, plantations or open areas, in general, are considered keystone structures to maintain biodiversity in landscapes, particular for forest organisms [67,68]. The composition of species in or around scattered trees is more similar to nearby forest areas than species composition in open areas where they are scattered, for a variety of organisms [68]. The size of scattered trees and landscape context will also affect how much of biodiversity and functional connectivity they may provide [69], but the main point is that much landscape biodiversity and ecosystem services it provides are secured by scattered trees in pastures and plantations.

Linear structures in open areas also provide pathways for organisms to cross distances in open areas between forest remnants, allowing functional connectivity in the landscape. Many crops are sown along rows, also known as 'row crops', such as manioc, sugarcane, corn and soybean in regions of the Atlantic Forest [70] and wheat in temperate regions [71], which provide a covered linear pathway for fast movement. Plantation rows may be oriented to connect forest remnants, or to direct animal movements towards trees or other linear structures in the matrix, such as hedgerows or live fences [72].

Corridors are indeed a more obvious choice for connectivity, but imply a large amount of investment, involved in land acquisition or large efforts to reforest a continuous strip of forest connecting remnants. Agroforestry systems connecting forest remnants may provide an alternative that will promote connectivity between populations of many species [73] and socioeconomic benefits for farmers or local residents. Agroforestry has the potential to reduce poverty and improve food security while addressing land degradation and supporting the delivery of other ecosystem services [74]. Trails along the AFT may also be connected by agrosytems or rural landscapes with biodiversity-friendly initiatives, promoting ecotourism.

Agroforestry systems do not need to provide structural, continuous vegetation connection between forest remnants, yet they may provide secondary habitat for some species, or stepping stones that allow

other species to cross the distance between forest remnants [75]. Stepping stones allow for functional connectivity between populations in forest remnants [76,77]. Even if not structurally connected, forest remnants may be functionally connected if immersed in a matrix of agropastoral activities that allow movement of organisms between forest remnants, such as cattle raising in silvipastoral systems [78]. Agroforest systems, through their hosting capacity, are not only able to maintain biodiversity in agricultural landscapes, but also help to consolidate or expand some forest fragments, further increasing landscape connectivity [79,80]. This is the key to a biodiversity friendly and sustainable land use. Functional connectivity between forest remnants reduces risks of regional extinction, maintaining ecosystem functioning and services it provides to humans, such as pollination, seed dispersal, nutrient cycling, carbon sequestration and biological control of pests in plantations [81–83].

The sustainability provided by agroforest systems may be especially relevant to connect parts of the AFT that currently are not structurally connected. Agroforest systems would provide not only scenic routes between forest fragments, but also biodiversity and ecosystem services to local farmers and residents. A meta-analysis for the Brazilian Atlantic Forest compared values of biodiversity and ecosystem services in different agroforestry systems with those found in conventional production systems and in old-growth forests. More diverse agroforestry systems had higher values of mean effect size for biodiversity and ecosystem service provision than simple agroforestry systems and conventional production systems [84].

Despite the relatively good state of conservation of the corridor along the AFT, if we look at the region between Desengano and Três Picos State Parks the scenario is different. This area is highly degraded but has high conservation importance to connect the two State Parks and several environmental protected areas (which in Brazilian system corresponds to IUCN category V), creating a biological corridor along the AFT. Most properties with environmental debt are in Nova Friburgo, followed by Santa Maria Madalena. While Nova Friburgo is a rich county, with a predominance of urbanized areas (houses and hotels) and pasture around the Três Picos State Park [34], Santa Maria Madalena is one of the poorest counties in the region. Landowners around the Desengano State Park produce mostly for subsistence or small local markets, and do not compete with other State production areas [32]. This may represent a great opportunity for forest restoration to comply with the law at lower cost, because the cost of restoring a productive land instead of using it for other practices will not be too high [23]. Forest restoration in the region has the opportunity of increasing landscape connectivity, promoting biodiversity conservation while increasing quality of life for local communities. Maintaining and restoring native vegetation in the area is important not only for conservation purposes, but also to promote ecosystem services, such as water and climate regulation, infiltration, water fertility and erosion protection. Meanwhile, cooperation between local population and the AFT has the potential to create opportunity for many small families that do not make a profit out of agriculture, benefiting from ecotourism while promoting biodiversity conservation [85]. This is important not only to increase the amount of forest around the trail and creating a great biological corridor, but also to straighten personal relations between the community and the State Parks. Local populations usually do not understand the importance of protected areas and their surroundings and usually see conservation as an impasse to economic development. At the Santa Maria Madalena County, only 40% of the population has visited the Desengano State Park, although 85% recognized that ecotourism linked to the park would have great positive impacts on society, creating job opportunities and increasing quality of life [85].

If ecotourism is an important revenue for local landowners, maintaining a good state of conservation is imperative to attract visitors [86]. Forest conservation and restoration within private lands in the area can give landowners the chance to comply with the law and benefit from the ecotourism linked to the AFT, diversifying landowners' income, conserving biodiversity and sustaining ecosystem services, such as erosion control and water quality. Similarly, a system of secondary trails along the AFT also need to be connected. Promoting connectivity between trails is promoting connectivity for a particular kind of humans, hikers. Connectivity for hikers also mean connectivity for a variety of organisms, for medium–large vertebrates, but also invertebrates, trees, epiphytes and herbs.

The survey of sustainable tourism showed that there are already several initiatives throughout the AFT that can be used as an example for other areas with similar characteristics. Thus, the trail can contribute to the improvement of tourism activity through the adoption of the best practices existing in other regions of the AFT itself. This result was already expected, since besides criteria related to connectivity between protected areas and forest remnants, among the guidelines used to define the route are the prioritization of passing through major tourist attractions, protected areas, local tourist routes or circuits and traditional communities that express interest in developing tourism [3]. The

objective was to strengthen existing local initiatives and also to spread the benefits of tourism from consolidated attractions to new, less popular areas. Taking into account that the details of the route in each section were defined in participatory meetings and surveys with the presence of volunteers and experts from each region, it was expected that they would have suggested passing through tourism initiatives considered exemplary.

The overlapping of the trail with local circuits or routes intends to promote the existing efforts aimed at different segments, such as rural, gastronomic or cultural tourism. Tourist routes are considered one of the most successful strategies for strengthening sustainable tourism [87]. An example is the project Pathways of the Southern Canyons Geopark, which mobilizes city halls and the tourist sector and proposes the recognition of a large part of the Serra Geral stretch as a geopark by UNESCO, an initiative to promote tourism associated with the valorization and landscape conservation [88]. Among the initiatives of rural tourism, we can mention the project *Acolhida na Colônia* (meaning welcome in the colony), in Santa Catarina state, which brings together small properties that work cooperatively in the production of organic food. The producers receive at their homes or in small chalets, integrating visitors into their daily activities.

The community-based tourism is considered an alternative for generating income and fixing populations in rural areas, especially when the initiatives come from the community itself [89]. Participation, power redistribution, collaboration processes and social capital formation, more than desirable objectives, are prerequisites for considering an initiative like community-based tourism. Along the AFT there are more than 50 traditional communities, including indigenous lands, quilombos (settlements founded by people of African origin) and *caiçara* communities (descendants from indigenous people, Europeans and Africans established in coastal areas). The *Quilombos do Ribeira* Circuit offers cultural experiences that involve getting to know the cuisine and agricultural techniques. In the Tenondé-Porã Indigenous Land, six Guarani villages are open for experiences of indigenous culture. Despite 500 years of colonization, this territory resists in the municipality of São Paulo, the largest city in Latin America [90].

The parks, in turn, show the great tourist potential of the trail. Among the attractions along the trail is the world-famous Corcovado and the Christ the Redeemer Statue, located in the Tijuca National Park, and some of the most iconic mountains, beaches and islands of Brazil. The significant proportion of the trail within parks is further evidence of its ecotourism potential, as they are the protected area category defined by their exceptional scenic beauty [36]. A strong evidence of the potential of the AFT was its assignment as one of the top 52 places to visit in 2020 by the *New York Times* (https://www.nytimes.com/interactive/2020/travel/places-to-visit.html).

# 5. Conclusion

Although the Atlantic Forest has been intensely devastated, the AFT still has a large part of its corridor conserved and the potential for recovery. The combined use of legal measures, such as the obligation to recover APPs and LR, and economic incentives to owners, with the promotion of agroforestry production and the development of tourism, can generate important results for conservation. By bringing together researchers working in a large geographical area and having volunteers working along the entire trail, AFT can also contribute to the structuring of a large-scale biodiversity and climate monitoring network. In this sense, by combining several initiatives, AFT can represent a significant change of scenery in the Serra do Mar corridor.

Data accessibility. All data used are described in tables 1 and 2 of the manuscript, and sources are homepage and reference since we do not used original data. Please follow the list of homepages and references: Aximoff *et al.* [37]. Lanna [28], http://caminhodamataatlantica.org.br/. Cid *et al.* [39], https://refauna.wixsite.com/site/projeto-anta. Eurico [40], http://www.institutomanaca.org.br/. Coles *et al.* [41], https://canionsdosul.org/. https://acolhida.com.br/, https://circuitovaleeuropeu.com.br/, http://grandereservamataatlantica.com.br/. https://www.quilombosdoribeira.org.br/, https://turismo.teresopolis.rj.gov.br/onde-ir/circuito-tere-fri/. Camargo [51], http://garupa.org.br/guia-garupa/maruja-ilha-do-cardoso/, https://tenondepora.org.br/, https://www.castelhanos.org/. https://www.otss.org.br/turismo-de-base-comunitaria, https://redenosdaguanabara.com.br.

Authors' contributions. C.E.V.G. conceived the idea, J.N. performed spatial analyses on restoration debt, E.B.V.C. compiled information on ecotourism, A.M.L. compiled information on biological monitoring, M.U. compiled information on agroforestry systems and its potential do corridors, M.V.V. compiled information on functional connectivity. C.E.V.G. took leadership in writing, and all authors contributed to the manuscript and gave final approval for publication.

Competing interests. We declare we have no competing interests.

Funding. This study was supported by CNPq, FAPERJ, INCT Ecologia, Evolução e Conservação da Biodiversidade (MCTIC/CNPq/FAPEG/465610/2014-5) and PPBio/CNPq/MCTIC.

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
