## [Peer Review File · Royal Society Open Science]

Review History

RSOS-201840.R0 (Original submission)

Review form: Reviewer 1

Is the manuscript scientifically sound in its present form?

Yes

Are the interpretations and conclusions justified by the results?

Yes

Is the language acceptable?

No

Do you have any ethical concerns with this paper?

No

Have you any concerns about statistical analyses in this paper?

No

Recommendation?

Major revision is needed (please make suggestions in comments)

Comments to the Author(s)

This study examines the land use and opportunities for forest restoration in the buffer zone (2 km on each side) of the Atlantic Forest Trail and discusses potential ways to restore degraded or deforested areas and increase connectivity of protected areas along the length of the corridor. It is not a hypothesis based paper, but more of a descriptive case study analysis of potential for restoration and for enhancing the role of this trail system in regional conservation and connectivity between the nearby protected areas. I enjoyed learning more about the trail. It would be great to have some photos of areas of the trail, iconic views, and potential for ecotourism. Also, it is important to point out that there is much more to the Atlantic Forest that is not included in the trail system. Currently, the trail system only cover areas in the south, beginning in Rio de Janeiro State. Is there a plan to extend this trail system northward??

Specific comments:

Page 4 line 28: in Brazil

Page 5 line 27: it would be nice to show a map here, indicating where the protected areas are and coding them for private vs. public protected areas.

Page 5 line 42: stability of what?

Page 5, line 45: just within the region of the trail? Throughout Atlantic Forest region? What is the scope of this compilation?

Page 5, line 60: bridges? not clear how the trail can go over the sea.

Page 6, line 16: please provide information on the resolution of these images (30 m). Also, how do you know this if forest cover and not tree cover (including plantations)?

Page 7 line 34 and 36: this is %LR in equation above. Need to be consistent in the abbreviation

Page 7, line 46: how is this region defined?

Page 8, line 25: this is an odd unit. Use ha or km². Also, can you calculate the increase in connectivity if the 2 km buffer zones were all reforested/restored?

Page 9, line 12: earlier you cite 13% (prior study). This estimate has changed based on recent studies using higher resolution imagery.

Crouzeilles, R., E. Santiami, M. Rosa, L. Pugliese, P. H. Brancalion, R. R. Rodrigues, J. P. Metzger, M. Calmon, C. A. d. M. Scaramuzza, and M. H. Matsumoto. 2019. There is hope for achieving ambitious Atlantic Forest restoration commitments. *Perspectives in ecology and conservation* 17:80-83.

Rezende, C., F. Scarano, E. Assad, C. Joly, J. Metzger, B. Strassburg, M. Tabarelli, G. Fonseca, and R. Mittermeier. 2018. From hotspot to hopespot: An opportunity for the Brazilian Atlantic Forest. *Perspectives in ecology and conservation* 16:208-214.

Page 9 line 51: also cite this new paper:

Chazdon, R. L., L. Cullen Jr, S. M. Padua, and C. Valladares Padua. 2020. People, primates, and predators in the Pontal: From endangered species conservation to forest and landscape restoration in Brazil's Atlantic Forest. *Royal Society Open Science* Chazdon, R. L., L. Cullen Jr, S. M. Padua, and C. V. Padua. 2020. People, primates and predators in the Pontal: from endangered

species conservation to forest and landscape restoration in Brazil's Atlantic Forest. Royal Society Open Science 7:200939.

Page 10, line 51: can you go further and propose specific restoration approaches for different regions of the trail?

Page 11, line 43: but smallholders do not have a way to finance restoration. They would need payments for environmental services to be able to restore forest. This could also jeopardize their food security.

Review form: Reviewer 2

Is the manuscript scientifically sound in its present form?

No

Are the interpretations and conclusions justified by the results?

No

Is the language acceptable?

No

Do you have any ethical concerns with this paper?

No

Have you any concerns about statistical analyses in this paper?

Yes

Recommendation?

Major revision is needed (please make suggestions in comments)

Comments to the Author(s)

The authors present an interesting study of the Atlantic Forest Trail within the Brazilian Atlantic forest. It appears that the authors investigate three main aspects of the trail based on a combination of geospatial analyses, literature review and surveys. The three aspects are 1) the amount of land that must be restored within the vicinity of the trails, 2) the biodiversity monitoring and 3) sustainable tourism. While I think the research is interesting, I personally find the MS difficult to follow, and I am unable to properly assess and determine the importance and quality of the work. I list down several sources of confusion that I hope the authors can use as a guide to further improve the manuscript, and view as constructive. Some of these points can be fixed simply by editing the language and phrasing of the MS.

- 1) The authors seem to use the term "native" and "natural" interchangeably when referencing vegetation. In ecology, "native" is typically used to refer to species that originate from that location, as opposed to the term "non-native" or "invasive" species.
- 2) The authors talk about plotting a 4km buffer around the trail, when the analyses use a 2 km buffer. 2km on each side of the trail means the buffer radius is 2km.
- 3) Listing out the research aims and hypotheses/research questions might clarify the purpose of this study.
- 4) LR and RL seems to be used interchangeably? Or is RL just a typo?
- 5) Where does the formula used to calculate debt come from? Is it based purely on the "Brazilian Native Vegetation Protection Law"? If so, the section here can use a more detailed description.

- 6) Debt here appears to be calculated in meters, but the spatial data resolution is in hectares. The spatial data might not be able to support the resolution required for the formula.
- 7) Restoration is also a very broad term. Do the authors mean natural regeneration of vegetation across multiple habitats? What about planted forests that can include non-native species?
- 8) There appears to be little information/statistics/new insights to be gained from the literature review of biodiversity monitoring work and tourism surveys besides where they are located. I think some sort of analyses would be beneficial. Maybe something like X% of the publications reviewed showed that biodiversity benefited? Or X% of the tourism projects surveyed was sustainable? Even if the authors are not comfortable with detailed statistical analyses, some descriptive statistics would help.
- 9) There are also discrepancies in the statistics reported in the results
- 10) Much of the discussion is not directly supported by the results, and it reads more like an opinion piece in its current form. For example, how does knowing 75% of the 2km buffer around the AFT help inform connectivity conservation? Many unanswered ecological questions still remain. E.g., will the protected habitats be large enough to be used by the faunal species?
- 11) Authors should avoid words such as “impressive diversification” and “opulence of the rainforest”, and instead rely on simple, objective and scientific language throughout the manuscript.

Decision letter (RSOS-201840.R0)

Dear Dr Grelle

The Editors assigned to your paper RSOS-201840 "Sustainability issues in a tropical mega trail" have now received comments from reviewers and would like you to revise the paper in accordance with the reviewer comments and any comments from the Editors. Please note this decision does not guarantee eventual acceptance.

Please submit your revised manuscript and required files (see below) no later than 21 days from today's (ie 06-Jan-2021) date. Note: the ScholarOne system will 'lock' if submission of the revision is attempted 21 or more days after the deadline. If you do not think you will be able to meet this deadline please contact the editorial office immediately.

Please note article processing charges apply to papers accepted for publication in Royal Society Open Science (<https://royalsocietypublishing.org/rsos/charges>). Charges will also apply to

papers transferred to the journal from other Royal Society Publishing journals, as well as papers submitted as part of our collaboration with the Royal Society of Chemistry (<https://royalsocietypublishing.org/rsos/chemistry>). Fee waivers are available but must be requested when you submit your revision (<https://royalsocietypublishing.org/rsos/waivers>).

on behalf of Dr Agnieszka Latawiec (Subject Editor)
openscience@royalsociety.org

Associate Editor Comments to Author (Dr Agnieszka Latawiec):

Associate Editor: 1

Comments to the Author:

Dear Authors

Please carefully incorporate the suggestions of both reviewers so the paper can be considered for publication in RSOS.

Reviewer comments to Author:

Reviewer: 1

Comments to the Author(s)

This study examines the land use and opportunities for forest restoration in the buffer zone (2 km on each side) of the Atlantic Forest Trail and discusses potential ways to restore degraded or deforested areas and increase connectivity of protected areas along the length of the corridor. It is not a hypothesis based paper, but more of a descriptive case study analysis of potential for restoration and for enhancing the role of this trail system in regional conservation and connectivity between the nearby protected areas. I enjoyed learning more about the trail. It would be great to have some photos of areas of the trail, iconic views, and potential for ecotourism. Also, it is important to point out that there is much more to the Atlantic Forest that is not included in the trail system. Currently, the trail system only cover areas in the south, beginning in Rio de Janeiro State. Is there a plan to extend this trail system northward??

Specific comments:

Page 4 line 28: in Brazil

Page 5 line 27: it would be nice to show a map here, indicating where the protected areas are and coding them for private vs. public protected areas.

Page 5 line 42: stability of what?

Page 5, line 45: just within the region of the trail? Throughout Atlantic Forest region? What is the scope of this compilation?

Page 5, line 60: bridges? not clear how the trail can go over the sea.

Page 6, line 16: please provide information on the resolution of these images (30 m). Also, how do you know this is forest cover and not tree cover (including plantations)?

Page 7 line 34 and 36: this is %LR in equation above. Need to be consistent in the abbreviation

Page 7, line 46: how is this region defined?

Page 8, line 25: this is an odd unit. Use ha or km². Also, can you calculate the increase in connectivity if the 2 km buffer zones were all reforested/restored?

Page 9, line 12: earlier you cite 13% (prior study). This estimate has changed based on recent studies using higher resolution imagery.

Crouzeilles, R., E. Santiami, M. Rosa, L. Pugliese, P. H. Brancalion, R. R. Rodrigues, J. P. Metzger, M. Calmon, C. A. d. M. Scaramuzza, and M. H. Matsumoto. 2019. There is hope for achieving ambitious Atlantic Forest restoration commitments. *Perspectives in ecology and conservation* 17:80-83.

Rezende, C., F. Scarano, E. Assad, C. Joly, J. Metzger, B. Strassburg, M. Tabarelli, G. Fonseca, and R. Mittermeier. 2018. From hotspot to hopespot: An opportunity for the Brazilian Atlantic Forest. *Perspectives in ecology and conservation* 16:208-214.

Page 9 line 51: also cite this new paper:

Chazdon, R. L., L. Cullen Jr, S. M. Padua, and C. Valladares Padua. 2020. People, primates, and predators in the Pontal: From endangered species conservation to forest and landscape restoration in Brazil's Atlantic Forest. *Royal Society Open Science* Chazdon, R. L., L. Cullen Jr, S. M. Padua, and C. V. Padua. 2020. People, primates and predators in the Pontal: from endangered species conservation to forest and landscape restoration in Brazil's Atlantic Forest. *Royal Society Open Science* 7:200939.

Page 10, line 51: can you go further and propose specific restoration approaches for different regions of the trail?

Page 11, line 43: but smallholders do not have a way to finance restoration. They would need payments for environmental services to be able to restore forest. This could also jeopardize their food security.

Reviewer: 2

Comments to the Author(s)

The authors present an interesting study of the Atlantic Forest Trail within the Brazilian Atlantic forest. It appears that the authors investigate three main aspects of the trail based on a combination of geospatial analyses, literature review and surveys. The three aspects are 1) the amount of land that must be restored within the vicinity of the trails, 2) the biodiversity monitoring and 3) sustainable tourism. While I think the research is interesting, I personally find the MS difficult to follow, and I am unable to properly assess and determine the importance and quality of the work. I list down several sources of confusion that I hope the authors can use as a guide to further improve the manuscript, and view as constructive. Some of these points can be fixed simply by editing the language and phrasing of the MS.

1) The authors seem to use the term "native" and "natural" interchangeably when referencing vegetation. In ecology, "native" is typically used to refer to species that originate from that location, as opposed to the term "non-native" or "invasive" species.

- 2) The authors talk about plotting a 4km buffer around the trail, when the analyses use a 2 km buffer. 2km on each side of the trail means the buffer radius is 2km.
- 3) Listing out the research aims and hypotheses/research questions might clarify the purpose of this study.
- 4) LR and RL seems to be used interchangeably? Or is RL just a typo?
- 5) Where does the formula used to calculate debt come from? Is it based purely on the “Brazilian Native Vegetation Protection Law”? If so, the section here can use a more detailed description.
- 6) Debt here appears to be calculated in meters, but the spatial data resolution is in hectares. The spatial data might not be able to support the resolution required for the formula.
- 7) Restoration is also a very broad term. Do the authors mean natural regeneration of vegetation across multiple habitats? What about planted forests that can include non-native species?
- 8) There appears to be little information/statistics/new insights to be gained from the literature review of biodiversity monitoring work and tourism surveys besides where they are located. I think some sort of analyses would be beneficial. Maybe something like X% of the publications reviewed showed that biodiversity benefited? Or X% of the tourism projects surveyed was sustainable? Even if the authors are not comfortable with detailed statistical analyses, some descriptive statistics would help.
- 9) There are also discrepancies in the statistics reported in the results
- 10) Much of the discussion is not directly supported by the results, and it reads more like an opinion piece in its current form. For example, how does knowing 75% of the 2km buffer around the AFT help inform connectivity conservation? Many unanswered ecological questions still remain. E.g., will the protected habitats be large enough to be used by the faunal species?
- 11) Authors should avoid words such as “impressive diversification” and “opulence of the rainforest”, and instead rely on simple, objective and scientific language throughout the manuscript.

===PREPARING YOUR MANUSCRIPT===

If you have been asked to revise the written English in your submission as a condition of publication, you must do so, and you are expected to provide evidence that you have received language editing support. The journal would prefer that you use a professional language editing service and provide a certificate of editing, but a signed letter from a colleague who is a native speaker of English is acceptable. Note the journal has arranged a number of discounts for authors

using professional language editing services
(<https://royalsociety.org/journals/authors/benefits/language-editing/>).

===PREPARING YOUR REVISION IN SCHOLARONE===

<https://royalsociety.org/journals/authors/author-guidelines/#supplementary-material> to include a suitable title and informative caption. An example of appropriate titling and captioning may be found at https://figshare.com/articles/Table_S2_from_Is_there_a_trade-

off_between_peak_performance_and_performance_breadth_across_temperatures_for_aerobic_sc
ope_in_teleost_fishes_/3843624.

Author's Response to Decision Letter for (RSOS-201840.R0)

See Appendix A.

Decision letter (RSOS-201840.R1)

Dear Dr Grelle,

It is a pleasure to accept your manuscript entitled "Sustainability issues in a tropical mega trail" in its current form for publication in Royal Society Open Science. The comments of the reviewer(s) who reviewed your manuscript are included at the foot of this letter.

You can expect to receive a proof of your article in the near future. Please contact the editorial office (openscience@royalsociety.org) and the production office (openscience_proofs@royalsociety.org) to let us know if you are likely to be away from e-mail contact – if you are going to be away, please nominate a co-author (if available) to manage the proofing process, and ensure they are copied into your email to the journal.

Kind regards,

Anita Kristiansen
Editorial Coordinator

on behalf of Agnieszka Latawiec (Subject Editor)
openscience@royalsociety.org

Appendix A

instituto de
biologia

UNIVERSIDADE
FEDERAL DO
RIO DE JANEIRO

Departamento de
ECOLOGIA

Rio de Janeiro, January, 24, 2021

Dear Dr Agnieszka Latawiec (Subject Editor),

Thank you very much for considering our manuscript to your journal. Please find below the answers for all the questions done by referees, and we appreciate their suggestions to clarify and improve the quality of our manuscript (ms). The answers are in bold.

Reviewer: 1

This study examines the land use and opportunities for forest restoration in the buffer zone (2 km on each side) of the Atlantic Forest Trail and discusses potential ways to restore degraded or deforested areas and increase connectivity of protected areas along the length of the corridor. It is not a hypothesis based paper, but more of a descriptive case study analysis of potential for restoration and for enhancing the role of this trail system in regional conservation and connectivity between the nearby protected areas. I enjoyed learning more about the trail. It would be great to have some photos of areas of the trail, iconic views, and potential for ecotourism.

Answer: We added eight photos as supplemental material.

Also, it is important to point out that there is much more to the Atlantic Forest that is not included in the trail system. Currently, the trail system only cover areas in the south, beginning in Rio de Janeiro State. Is there a plan to extend this trail system northward??

Answer: There isn't a plan to extend the Atlantic Forest trail (AFT). As argued in lines 19 and 20 of page 5 in the ms, the AFT was "inspired in the North-American Apalachian Trail, in 2012 was launched the idea of implementing a mega trail crossing the Serra do Mar and Serra Geral mountain ranges". Thus, in essence there is a geographic limit to AFT.

Specific comments:

Page 4 line 28: in Brazil

Answer: Done.

Page 5 line 27: it would be nice to show a map here, indicating where the protected areas are and coding them for private vs. public protected areas.

Answer: Done. We prepared a new figure with private vs. public protected areas, which will correspond to figure 1 in the new version.

Page 5 line 42: stability of what?

Answer: Stability of percentage of species whose home-range fit in a corridor of a given width. Thus, according to a key-reference in this subject (Beier 2009), an asymptote exists when corridor width reaches 4km. We added additional details in the text to clarify this point.

Page 5, line 45: just within the region of the trail? Throughout Atlantic Forest region? What is the scope of this compilation?

Answer: Initiatives of tourism and biodiversity monitoring were compiled just within the region of the trail. The essence of our ms is to describe the socio-ecological potential of Atlantic Forest Trail.

Page 5, line 60: bridges? not clear how the trail can go over the sea.

Answer: Boats will be necessary to cross the sea in some parts of the Atlantic Forest Trail. We added this detail into the text.

Page 6, line 16: please provide information on the resolution of these images (30 m). Also, how do you know this if forest cover and not tree cover (including plantations)?

Answer: You are right and we added the resolution (30m) in the text. Also, MapBiomass Project gives different legends to native forest, pasturelands, mosaic of agro-pastoral land uses, monoculture tree plantation and croplands.

Page 7 line 34 and 36: this is %LR in equation above. Need to be consistent in the abbreviation

Answer: Yes and thank you for show this mistake. We changed all to LR%.

Page 7, line 46: how is this region defined?

Answer: The region is the Atlantic Forest Trail plus a buffer with 4km of width. We added this information into the text.

Page 8, line 25: this is an odd unit. Use ha or km². Also, can you calculate

the increase in connectivity if the 2 km buffer zones were all reforested/restored?

Answer: You are right and we changed it to km².

Analyses of connectivity is a subject of other ms, and we didn't finish the analyses yet.

Page 9, line 12: earlier you cite 13% (prior study). This estimate has changed based on recent studies using higher resolution imagery.

Answer: A recent paper found a cover of native forest of 13% in Atlantic Forest (Souza et al. 2020) and we follow it along the text.

Page 9 line 51: also cite this new paper:

Chazdon, R. L., L. Cullen Jr, S. M. Padua, and C. V. Padua. 2020. People, primates and predators in the Pontal: from endangered species conservation to forest and landscape restoration in Brazil's Atlantic Forest. Royal Society Open Science 7:200939.

Answer: Okay, and thank you. We didn't know this new paper that is strictly related with our ms.

Page 10, line 51: can you go further and propose specific restoration approaches for different regions of the trail?

Answer: This is very difficult, and we need to study in detail each region. Furthermore, the main deforested gap between PED and PETP. For other regions apparently there isn't a great restoration debt .

Page 11, line 43: but smallholders do not have a way to finance restoration. They would need payments for environmental services to be able to restore forest. This could also jeopardize their food security.

Answer: Yes. A lack of specific budget is a problem to food security, but we didn't have the intention to explore this in our ms. In reality, this an important task that deserves a new study and paper.

Reviewer: 2

Comments to the Author(s)

The authors present an interesting study of the Atlantic Forest Trail within the Brazilian Atlantic forest. It appears that the authors investigate three main aspects of the trail based on a combination of geospatial analyses, literature review and surveys. The three aspects are 1) the amount of land that must be restored within the vicinity of the trails, 2) the biodiversity

monitoring and 3) sustainable tourism. While I think the research is interesting, I personally find the MS difficult to follow, and I am unable to properly assess and determine the importance and quality of the work. I list down several sources of confusion that I hope the authors can use as a guide to further improve the manuscript, and view as constructive. Some of these points can be fixed simply by editing the language and phrasing of the MS.

1) The authors seem to use the term "native" and "natural" interchangeably when referencing vegetation. In ecology, "native" is typically used to refer to species that originate from that location, as opposed to the term "non-native" or "invasive" species.

Answer: Thank you and you are right. We changed it to "native" along the text.

2) The authors talk about plotting a 4km buffer around the trail, when the analyses use a 2 km buffer. 2km on each side of the trail means the buffer radius is 2km.

Answer: We performed analyses with complete buffer of 4km (2km on each side equal 4km).

3) Listing out the research aims and hypotheses/research questions might clarify the purpose of this study.

Answer: We added aims in the text. Also, it is impossible to get hypotheses now, and we are just compiling the socio-ecological potential of Atlantic Forest Trail. Reviewer #1 understood very well the limits and merits of our ms: "It is not a hypothesis based paper, but more of a descriptive case study analysis of potential for restoration and for enhancing the role of this trail system in regional conservation and connectivity between the nearby protected areas."

4) LR and RL seems to be used interchangeably? Or is RL just a typo?

Answer: Legal Reserve (LR) is the correct term, and we changed all RL to LR.

5) Where does the formula used to calculate debt come from? Is it based purely on the "Brazilian Native Vegetation Protection Law"? If so, the section here can use a more detailed description.

Answer: The formula of restoration debt was obtained in Niemeyer et al (2020) and it is based on requirements of Brazilian Native Vegetation Protection Law.

6) Debt here appears to be calculated in meters, but the spatial data

resolution is in hectares. The spatial data might not be able to support the resolution required for the formula.

Answer: Sorry, but I didn't understand the point, since MapBiomas Project, used in the analyses of this ms, has a resolution in meters.

7) Restoration is also a very broad term. Do the authors mean natural regeneration of vegetation across multiple habitats? What about planted forests that can include non-native species?

Answer: Natural regeneration is an option in many parts of Atlantic Forest, as showed by Crouzeilles et al (2021) but it demands a long time. The plan to the restoration between PED and PETP parks is to use native plants to accelerate the connectivity between these two parks.

8) There appears to be little information/statistics/new insights to be gained from the literature review of biodiversity monitoring work and tourism surveys besides where they are located. I think some sort of analyses would be beneficial. Maybe something like X% of the publications reviewed showed that biodiversity benefited? Or X% of the tourism projects surveyed was sustainable? Even if the authors are not comfortable with detailed statistical analyses, some descriptive statistics would help.

Answer: Thank you for your comment. We understand the suggestion of including some sort of analyses if this was a review. However, rather than an exhaustive research, the aim of this part was to map selection of initiatives that can be replicated and implemented in other regions of the biome. This was not clear in the first version of the ms and we have now re written this part.

9) There are also discrepancies in the statistics reported in the results

Answer: We checked and corrected it. Thank you.

10) Much of the discussion is not directly supported by the results, and it reads more like an opinion piece in its current form. For example, how does knowing 75% of the 2km buffer around the AFT help inform connectivity conservation? Many unanswered ecological questions still remain. E.g., will the protected habitats be large enough to be used by the faunal species?

Answer: We chose a buffer with 2km on each side (total of 4km) following a key-reference on the minimum buffer width to allow occurrence and flow of individuals of fauna species (Beier 2009).

We added additional details in the text to clarify.

Furthermore, the ongoing studies on monitoring of species occurrence, some listed in the text and others only planned and waiting for budget, will respond the question if protected habitats are large enough to be used by the faunal species.

11) Authors should avoid words such as “impressive diversification” and “opulence of the rainforest”, and instead rely on simple, objective and scientific language throughout the manuscript.

Answer: We agree, and changed the text deleting words such as impressive and opulence.

Reference cited:

Beier P. 2019. A rule of thumb for widths of conservation corridors. *Conservation Biology*, **33**, 976–978.

Crouzeilles, R.; Beyer, H.L.; Monteiro, L.M.; Feltran-Barbieri, R.; Pessôa, A.C.M.; Barros, F.S.M.; Lindenmayer, D.B.; Lino, E.D.S.M.; Grelle, C.E.V.; Chazdon, R.L.; et al. Achieving cost-effective landscape-scale forest restoration through targeted natural regeneration. *Conserv. Lett.* 2020, 2020, e12709.

Niemeyer J, Barros FS, Silva DS, Crouzeilles R, Vale MM. 2020. Planning forest restoration within private land holdings with conservation co-benefits at the landscape scale. *Science of The Total Environment*, **717**, 135262. <https://doi.org/10.1016/j.scitotenv.2019.135262>

Souza Jr., C.M., Shimbo, Julia Z., Rosa, M.R., Parente, L.L., Alencar, A.A., Rudorff, B.F.T., Hasenack, H., Matsumoto, M., Ferreira, L.G., Souza-Filho, P.W.M., de Oliveira, S.W., Rocha, W.F., Fonseca, A.V., Marques, C.B., Diniz, C.G., Costa, D., Monteiro, D., Rosa, E.R., Velez-Martin, E., Weber, E.J., Lenti, F.E.B., Paternost, F.F., Pareyn, F.G. C., Siqueira, J.V., Viera, J.L., Neto, L.C.F., Saraiva, M.M., Sales, M.H., Salgado, M.P. G., Vasconcelos, R., Galano, S., Mesquita, V.V., Azevedo, T., 2020. Reconstructing three decades of land use and land cover changes in Brazilian biomes with Landsat archive and earth engine. *Remote Sens.* 12, 2735

Sincerely,

Carlos E V Grelle, PhD (on behalf of all authors)